# Efficient Optimization for Average Precision SVM

**Pritish Mohapatra**
IIIT Hyderabad
pritish.mohapatra@research.iiit.ac.in

**C.V. Jawahar**
IIIT Hyderabad
jawahar@iiit.ac.in

**M. Pawan Kumar**
Ecole Centrale Paris & INRIA Saclay
pawan.kumar@ecp.fr

## Abstract

The accuracy of information retrieval systems is often measured using average precision (AP). Given a set of positive (relevant) and negative (non-relevant) samples, the parameters of a retrieval system can be estimated using the AP-SVM framework, which minimizes a regularized convex upper bound on the empirical AP loss. However, the high computational complexity of loss-augmented inference, which is required for learning an AP-SVM, prohibits its use with large training datasets. To alleviate this deficiency, we propose three complementary approaches. The first approach guarantees an asymptotic decrease in the computational complexity of loss-augmented inference by exploiting the problem structure. The second approach takes advantage of the fact that we do not require a full ranking during loss-augmented inference. This helps us to avoid the expensive step of sorting the negative samples according to their individual scores. The third approach approximates the AP loss over all samples by the AP loss over difficult samples (for example, those that are incorrectly classified by a binary SVM), while ensuring the correct classification of the remaining samples. Using the PASCAL VOC action classification and object detection datasets, we show that our approaches provide significant speed-ups during training without degrading the test accuracy of AP-SVM.

## 1  Introduction

Information retrieval systems require us to rank a set of samples according to their relevance to a query. The parameters of a retrieval system can be estimated by minimizing the prediction risk on a training dataset, which consists of positive and negative samples. Here, positive samples are those that are relevant to a query, and negative samples are those that are not relevant to the query. Several risk minimization frameworks have been proposed in the literature, including structured support vector machines (SSVM) [15, 16], neural networks [14], decision forests [11] and boosting [13]. In this work, we focus on SSVMs for clarity while noting the methods we develop are also applicable to other learning frameworks.

The SSVM framework provides a linear prediction rule to obtain a structured output for a structured input. Specifically, the score of a putative output is the dot product of the parameters of an SSVM with the joint feature vector of the input and the output. The prediction requires us to maximize the score over all possible outputs for an input. During training, the parameters of an SSVM are estimated by minimizing a regularized convex upper bound on a user-specified loss function. The loss function measures the prediction risk, and should be chosen according to the evaluation criterion for the system. While in theory the SSVM framework can be employed in conjunction with any loss function, in practice its feasibility depends on the computational efficiency of the corresponding *loss-augmented inference*. In other words, given the current estimate of the parameters, it is important to be able to efficiently maximize the sum of the score and the loss function over all possible outputs.

A common measure of accuracy for information retrieval is average precision (AP), which is used in several standard challenges such as the PASCAL VOC object detection, image classification and action classification tasks [7], and the TREC Web Track corpora. The popularity of AP inspired Yue *et al.* [19] to propose the AP-SVM framework, which is a special case of SSVM. The input of AP-SVM is a set of samples, the output is a ranking and the loss function is one minus the AP of the ranking. In order to learn the parameters of an AP-SVM, Yue *et al.* [19] developed an optimal greedy algorithm for loss-augmented inference. Their algorithm consists of two stages. First, it sorts the positive samples $\mathcal{P}$ and the negative samples $\mathcal{N}$ separately in descending order of their individual scores. The individual score of a sample is equal to the dot product of the parameters with the feature vector of the sample. Second, starting from the negative sample with the highest score, it iteratively finds the optimal interleaving rank for each of the $|\mathcal{N}|$ negative samples. The interleaving rank for a negative sample is the index of the highest ranked positive sample ranked below it. which requires at most $O(|\mathcal{P}|)$ time per iteration. The overall algorithm is described in detail in the next section. Note that, typically $|\mathcal{N}| \gg |\mathcal{P}|$, that is, the negative samples significantly outnumber the positive samples.

While the AP-SVM has been successfully applied for ranking using high-order information in mid to large size datasets [5], many methods continue to use the simpler binary SVM framework for large datasets. Unlike AP-SVM, a binary SVM optimizes the surrogate 0-1 loss. Its main advantage is the efficiency of the corresponding loss-augmented inference algorithm, which has a complexity of $O(|\mathcal{P}| + |\mathcal{N}|)$. However, this gain in training efficiency often comes at the cost of a loss in testing accuracy, which is especially significant when training with weakly supervised datasets [1].

In order to facilitate the use of AP-SVM, we present three complementary approaches to speed-up its learning. Our first approach exploits an interesting structure in the problem corresponding to the computation of the rank of the $j$-th negative sample. Specifically, we show that when $j > |\mathcal{P}|$, the rank of the $j$-th negative sample is obtained by maximizing a discrete *unimodal function*. Here, a discrete function defined over points $\{1, \cdots, p\}$ is said to be unimodal if it is non-decreasing from $\{1, \cdots, k\}$ and non-increasing from $\{k, \cdots, p\}$ for some $k \in \{1, \cdots, p\}$. Since the mode of a discrete unimodal function can be computed efficiently using binary search, it reduces the computational complexity of computing the rank of the $j$-th negative sample from $O(|\mathcal{P}|)$ to $O(\log(|\mathcal{P}|))$. To the best of our knowledge, ours is the first work to improve the speed of loss-augmented inference for AP-SVM by taking advantage of the special structure of the problem. Unlike [2] which proposes an efficient method for a similar framework of structured output ranking, our method optimizes the APloss.

Our second approach relies on the fact that in many cases we do not need to explicitly compute the optimal interleaving rank for all the negative samples. Specifically, we only need to compute the interleaving rank for the set of negative samples that would have an interleaving rank of less than $|\mathcal{P}| + 1$. We identify this set using a binary search over the list of negative samples. While training, after the initial few training iterations the size of this set rapidly reduces, allowing us to significantly reduce the training time in practice.

Our third approach uses the intuition that the 0-1 loss and the AP loss differ only when some of the samples are difficult to classify (that is, some positive samples that can be confused as negatives and vice versa). In other words, when the 0-1 loss over the training dataset is 0, then the AP loss is also 0. Thus, instead of optimizing the AP loss over all the samples, we adopt a two-stage approximate strategy. In the first stage, we identify a subset of difficult samples (specifically, those that are incorrectly classified by a binary SVM). In the second stage, we optimize the AP loss over the subset of difficult samples, while ensuring the correct classification of the remaining easy samples. Using the PASCAL VOC action classification and object detection datasets, we empirically demonstrate that each of our approaches greatly reduces the training time of AP-SVM while not decreasing the testing accuracy.

## 2   The AP-SVM Framework

We provide a brief overview of the AP-SVM framework, highlighting only those aspects that are necessary for the understanding of this paper. For a detailed description, we refer the reader to [19].

**Input and Output.**   The input of an AP-SVM is a set of $n$ samples, which we denote by $\mathbf{X} = \{\mathbf{x}_i, i = 1, \cdots, n\}$. Each sample can either belong to the positive class (that is, the sample is

relevant) or the negative class (that is, the sample is not relevant). The indices for the positive and negative samples are denoted by $\mathcal{P}$ and $\mathcal{N}$ respectively. In other words, if $i \in \mathcal{P}$ and $j \in \mathcal{N}$ then $\mathbf{x}_i$ belongs to positive class and $\mathbf{x}_j$ belongs to the negative class.

The desired output is a ranking matrix $\mathbf{R}$ of size $n \times n$, such that (i) $\mathbf{R}_{ij} = 1$ if $\mathbf{x}_i$ is ranked higher than $\mathbf{x}_j$; (ii) $\mathbf{R}_{ij} = -1$ if $\mathbf{x}_i$ is ranked lower than $\mathbf{x}_j$; and (iii) $\mathbf{R}_{ij} = 0$ if $\mathbf{x}_i$ and $\mathbf{x}_j$ are assigned the same rank. During training, the ground-truth ranking matrix $\mathbf{R}^*$ is defined as: (i) $\mathbf{R}_{ij}^* = 1$ and $\mathbf{R}_{ji}^* = -1$ for all $i \in \mathcal{P}$ and $j \in \mathcal{N}$; (ii) $\mathbf{R}_{ii'}^* = 0$ and $\mathbf{R}_{jj'}^* = 0$ for all $i, i' \in \mathcal{P}$ and $j, j' \in \mathcal{N}$.

**Joint Feature Vector.**   For a sample $\mathbf{x}_i$, let $\psi(\mathbf{x}_i)$ denote its feature vector. The joint feature vector of the input $\mathbf{X}$ and an output $\mathbf{R}$ is specified as

$$\Psi(\mathbf{X}, \mathbf{R}) = \frac{1}{|\mathcal{P}||\mathcal{N}|} \sum_{i \in \mathcal{P}} \sum_{j \in \mathcal{N}} \mathbf{R}_{ij}(\psi(\mathbf{x}_i) - \psi(\mathbf{x}_j)). \tag{1}$$

In other words, the joint feature vector is the scaled sum of the difference between the features of all pairs of samples, where one sample is positive and the other is negative.

**Parameters and Prediction.**   The parameter vector of AP-SVM is denoted by $\mathbf{w}$, and is of the same size as the joint feature vector. Given the parameters $\mathbf{w}$, the ranking of an input $\mathbf{X}$ is predicted by maximizing the score, that is,

$$\mathbf{R} = \underset{\overline{\mathbf{R}}}{\operatorname{argmax}} \, \mathbf{w}^\top \Psi(\mathbf{X}, \overline{\mathbf{R}}). \tag{2}$$

Yue *et al.* [19] showed that the above optimization can be performed efficiently by sorting the samples $\mathbf{x}_k$ in descending order of their individual scores, that is, $s_k = \mathbf{w}^\top \psi(\mathbf{x}_k)$.

**Parameter Estimation.**   Given the input $\mathbf{X}$ and the ground-truth ranking matrix $\mathbf{R}^*$, we estimate the AP-SVM parameters by optimizing a regularized upper bound on the empirical AP loss. The AP loss of an output $\mathbf{R}$ is defined as $1 - \text{AP}(\mathbf{R}^*, \mathbf{R})$, where $\text{AP}(\cdot, \cdot)$ corresponds to the AP of the ranking $\mathbf{R}$ with respect to the true ranking $\mathbf{R}^*$. Specifically, the parameters are obtained by solving the following convex optimization problem:

$$\min_{\mathbf{w}} \quad \frac{1}{2}||\mathbf{w}||^2 + C\xi, \tag{3}$$
$$\text{s.t.} \quad \mathbf{w}^\top \Psi(\mathbf{X}, \mathbf{R}^*) - \mathbf{w}^\top \Psi(\mathbf{X}, \mathbf{R}) \geq \Delta(\mathbf{R}^*, \mathbf{R}) - \xi, \forall \mathbf{R}$$

The computational complexity of solving the above problem depends on the complexity of the corresponding loss-augmented inference, that is,

$$\hat{\mathbf{R}} = \underset{\overline{\mathbf{R}}}{\operatorname{argmax}} \, \mathbf{w}^\top \Psi(\mathbf{X}, \overline{\mathbf{R}}) + \Delta(\mathbf{R}^*, \overline{\mathbf{R}}). \tag{4}$$

For a given set of parameters $\mathbf{w}$, the above problem requires us to find the most violated ranking, that is, the ranking that maximizes the sum of the score and the AP loss. To be more precise, what we require is the joint feature vector $\Psi(\mathbf{X}, \hat{\mathbf{R}})$ and the AP loss $\Delta(\mathbf{R}^*, \hat{\mathbf{R}})$ corresponding to the most violated ranking. Yue *et al.* [19] provided an optimal greedy algorithm for problem (4), which is summarized in Algorithm 1. It consists of two stages. First, it sorts the positive and the negative samples separately in descending order of their scores (steps 1-2). This takes $O(|\mathcal{P}|\log(|\mathcal{P}|) + |\mathcal{N}|\log(|\mathcal{N}|))$ time. Second, starting with the highest scoring negative sample, it iteratively finds the interleaving rank of each negative sample $\mathbf{x}_j$. This involves maximizing the quantity $\delta_j(i)$, defined in equation (5), over all $i \in \{1, \cdots, |\mathcal{P}|\}$ (steps 3-7), which takes $O(|\mathcal{P}||\mathcal{N}|)$ time.

## 3 Efficient Optimization for AP-SVM

In this section, we propose three methods to speed up the training procedure of AP-SVM. The first two methods are exact. Specifically, they reduce the time taken to perform loss-augmented inference while ensuring the computation of the same most violated ranking as Algorithm 1. The third method provides a framework for a sensible trade-off between training efficiency and test accuracy.

### 3.1 Efficient Search for Loss-Augmented Inference

In order to find the most violated ranking, the greedy algorithm of Yue *et al.* [19] iteratively computes the optimal interleaving rank $opt_j \in \{1, \cdots, |\mathcal{P}| + 1\}$ for each negative sample $\mathbf{x}_j$ (step 5

**Algorithm 1** *The optimal greedy algorithm for loss-augmented inference for training* AP-SVM.

**input** Training samples $\mathbf{X}$ containing positive samples $\mathcal{P}$ and negative samples $\mathcal{N}$, parameters $\mathbf{w}$.
1: Sort the positive samples in descending order of the scores $s_i^p = \mathbf{w}^\top \psi(\mathbf{x}_i)$, $i \in \{1, \ldots, |\mathcal{P}|\}$.
2: Sort the negative samples in descending order of the scores $s_j^n = \mathbf{w}^\top \psi(\mathbf{x}_j)$, $j \in \{1, \ldots, |\mathcal{N}|\}$.
3: Set $j = 1$.
4: **repeat**
5:   Compute the interleaving rank $opt_j = \mathrm{argmax}_{i \in \{1, \cdots, |\mathcal{P}|\}} \, \delta_j(i)$, where

$$\delta_j(i) = \sum_{k=i}^{|\mathcal{P}|} \left\{ \frac{1}{|\mathcal{P}|} \left( \frac{j}{j+k} - \frac{j-1}{j+k-1} \right) - \frac{2(s_k^p - s_j^n)}{|\mathcal{P}||\mathcal{N}|} \right\}. \tag{5}$$

   The $j$-th negative sample is ranked between the $(opt_j - 1)$-th and the $opt_j$-th positive sample.
6:   Set $j \leftarrow j + 1$.
7: **until** $j > |\mathcal{N}|$.

---

of Algorithm 1). The interleaving rank $opt_j$ specifies that the negative sample $\mathbf{x}_j$ must be ranked between the $(opt_j - 1)$-th and the $opt_j$-th positive sample. The computation of the optimal interleaving rank for a particular negative sample requires us to maximize the discrete function $\delta_j(i)$ over the domain $i \in \{1, \cdots, |\mathcal{P}|\}$. Yue *et al.* [19] use a simple linear algorithm for this step, which takes $O(|\mathcal{P}|)$ time. In contrast, we propose a more efficient algorithm to maximize $\delta_j(\cdot)$, which exploits the special structure of this discrete function.

Before we describe our efficient algorithm in detail, we require the definition of a unimodal function. A discrete function $f : \{1, \cdots, p\} \leftarrow \mathbb{R}$ is said to be unimodal if and only if there exists a $k \in \{1, \cdots, p\}$ such that

$$\begin{aligned} f(i) \leq f(i+1), &\forall i \in \{1, \cdots, k-1\}, \\ f(i-1) \geq f(i), &\forall i \in \{k+1, \cdots, p\}. \end{aligned} \tag{6}$$

In other words, a unimodal discrete function is monotonically non-decreasing in the interval $[1, k]$ and monotonically non-increasing in the interval $[k, p]$. The maximization of a unimodal discrete function over its domain $\{1, \cdots, p\}$ simply requires us to find the index $k$ that satisfies the above properties. The maximization can be performed efficiently, in $O(\log(p))$ time, using binary search.

We are now ready to state the main result that allows us to compute the optimal interleaving rank of a negative sample efficiently.

**Proposition 1.** *The discrete function $\delta_j(i)$, defined in equation (5), is unimodal in the domain $\{1, \cdots, p\}$, where $p = \min\{|\mathcal{P}|, j\}$.*

The proof of the above proposition is provided in Appendix A (supplementary material).

---

**Algorithm 2** *Efficient search for the optimal interleaving rank of a negative sample.*

**input** $\{\delta_j(i), i = 1, \cdots, |\mathcal{P}|\}$.
1: $p = \min\{|\mathcal{P}|, j\}$.
2: Compute an interleaving rank $i_1$ as

$$i_i = \underset{i \in \{1, \cdots, p\}}{\mathrm{argmax}} \, \delta_j(i). \tag{7}$$

3: Compute an interleaving rank $i_2$ as

$$i_2 = \underset{i \in \{p+1, \cdots, |\mathcal{P}|\}}{\mathrm{argmax}} \, \delta_j(i). \tag{8}$$

4: Compute the optimal interleaving rank $opt_j$ as

$$opt_j = \begin{cases} i_1 & \text{if } \delta_j(i_1) \geq \delta_j(i_2), \\ i_2 & \text{otherwise.} \end{cases} \tag{9}$$

---

Using the above proposition, the discrete function $\delta_j(i)$ can be optimized over the domain $\{1, \cdots, |\mathcal{P}|\}$ efficiently as described in Algorithm 2. Briefly, our efficient search algorithm finds an interleaving ranking $i_1$ over the domain $\{1, \cdots, p\}$, where $p$ is set to $\min\{|\mathcal{P}|, j\}$ in order to ensure that the function $\delta_j(\cdot)$ is unimodal (step 2 of Algorithm 2). Since $i_1$ can be computed using binary search, the computational complexity of this step is $O(\log(p))$. Furthermore, we find an interleaving ranking $i_2$ over the domain $\{p + 1, \cdots, |\mathcal{P}|\}$ (step 3 of Algorithm 2). Since $i_2$ needs to be computed using linear search, the computational complexity of this step is $O(|\mathcal{P}| - p)$ when $p < |\mathcal{P}|$ and 0 otherwise. The optimal interleaving ranking $opt_j$ of the negative sample $\mathbf{x}_j$ can then be computed by comparing the values of $\delta_j(i_1)$ and $\delta_j(i_2)$ (step 4 of Algorithm 2).

Note that, in a typical training dataset, the negative samples significantly outnumber the positive samples, that is, $|\mathcal{N}| \gg |\mathcal{P}|$. For all the negative samples $\mathbf{x}_j$ where $j \geq |\mathcal{P}|$, $p$ will be equal to $|\mathcal{P}|$. Hence, the maximization of $\delta_j(\cdot)$ can be performed efficiently over the entire domain $\{1, \cdots, |\mathcal{P}|\}$ using binary search in $O(\log(|\mathcal{P}|))$ as opposed to the $O(|\mathcal{P}|)$ time suggested in [19].

## 3.2 Selective Ranking for Loss-Augmented Inference

While the efficient search algorithm described in the previous subsection allows us to find the optimal interleaving rank for a particular negative sample, the overall loss-augmented inference would still remain computationally inefficient when the number of negative samples is large (as is typically the case). This is due to the following two reasons. First, loss-augmented inference spends a considerable amount of time sorting the negative samples according to their individual scores (step 2 of Algorithm 1). Second, if we were to apply our efficient search algorithm to every negative sample, the total computational complexity of the second stage of loss-augmented inference (step 3-7 of Algorithm 1) will still be $O(|\mathcal{P}|^2 + (|\mathcal{N}| - |\mathcal{P}|) \log(|\mathcal{P}|))$.

In order to overcome the above computational issues, we exploit two key properties of loss-augmented inference in AP-SVM. First, if a negative sample $\mathbf{x}_j$ has the optimal interleaving rank $opt_j = |\mathcal{P}| + 1$, then all the negative samples that have lower score than $\mathbf{x}_j$ would also have the same optimal interleaving rank (that is, $opt_k = opt_j = |\mathcal{P}| + 1$ for all $k > j$). This property follows directly from the analysis of Yue *et al.* [19] who showed that, for $k < j$, $opt_k \geq opt_j$ and for any negative sample $\mathbf{x}_j$, $opt_j \in [1, |\mathcal{P}| + 1]$. We refer the reader to [19] for a detailed proof. Second, we note that the desired output of loss-augmented inference is not the most violated ranking $\hat{\mathbf{R}}$, but the joint feature vector $\Psi(\mathbf{X}, \hat{\mathbf{R}})$ and the AP loss $AP(\mathbf{R}^*, \hat{\mathbf{R}})$. From the definition of the joint feature vector and the AP loss, it follows that they do not depend on the relative ranking of the negative samples that share the same optimal interleaving rank. Specifically, both the joint feature vector and the AP loss only depend on the number of negatives that are ranked higher and lower than each positive sample.

The above two observations suggest the following alternate strategy to Algorithm 1. Instead of explicitly computing the optimal interleaving rank for each negative sample (which can be computationally expensive), we compute it only for negative samples that are expected to have optimal interleaving rank less than $|\mathcal{P}| + 1$. Algorithm 3 outlines the procedure we propose in detail. We first find the score $\hat{s}$ such that every negative sample $\mathbf{x}_j$ with score $s_j^n < \hat{s}$ has $opt_j = |\mathcal{P}| + 1$. We do a binary search over the list of scores of negative samples to find $\hat{s}$ (step 4 of algorithm 3). We do not need to sort the scores of all the negative samples, as we use the quick select algorithm to find the $k$-th highest score wherever required.

If the output of the loss-augmented inference is such that a large number of negative samples have optimal interleaving rank as $|\mathcal{P}| + 1$, then this alternate strategy would result in a significant speed-up during training. In our experiments, we found that in later iterations of the optimization, this is indeed the case in practice. Figure 1 shows how the number of negative samples with optimal interleaving rank equal to $|\mathcal{P}| + 1$, rapidly increases after

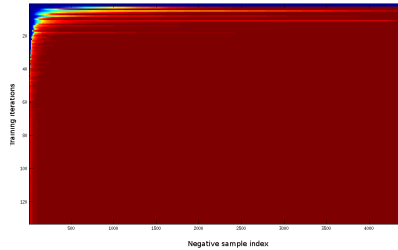

Figure 1: *A row corresponds to the interleaving ranks of the negative samples after a training iteration. Here, there are 4703 negative samples, and 131 training iterations. The interleaving ranks are represented using a heat map where the deepest red represents interleaving rank of $|\mathcal{P}| + 1$. (The figure is best viewed in colour.)*

---

**Algorithm 3** *The selective ranking algorithm for loss-augmented inference in* AP-SVM.

---
**input** $S^x$, $S^{\bar{x}}$, $|\mathcal{P}|$, $|\mathcal{N}|$
 1: Sort the positive samples in descending order of their scores $S^x$.
 2: Do binary search over $S^{\bar{x}}$ to find $\hat{s}$.
 3: Set $\mathcal{N}_l = \left\{ j \in \mathcal{N} | s_j^n < \hat{s} \right\}$
 4: Sort $\mathcal{N}_l$ in descending order of the scores.
 5: **for all** $j \in \mathcal{N}_l$ **do**
 6:     Compute $opt_j$ using Algorithm 2.
 7: **end for**
 8: Set $\mathcal{N}_r = \mathcal{N} - \mathcal{N}_l$.
 9: **for all** $j \in \mathcal{N}_r$ **do**
10:     Set $opt_j = |\mathcal{P}| + 1$.
11: **end for**
**output** $opt_j$ , $\forall j \in \mathcal{N}$

---

a few training iterations for a typical experiment. A large number of negative samples have optimal interleaving rank equal to $|\mathcal{P}| + 1$, while the negative samples that have other values of optimal interleaving rank decrease considerably.

It would be worth taking note that here, even though we take advantage of the fact that a long sequence of negative samples at the end of the list take the same optimal interleaving rank, such sequences also occur at other locations throughout the list. This can be leveraged for further speed-up by computing the interleaving rank for only the boundary samples of such sequences and setting all the intermediate samples to the same interleaving rank as the boundary samples. We can use a method similar to the one presented in this section to search for such sequences by using the quick select algorithm to compute the interleaving rank for any particular negative sample on the list.

### 3.3 Efficient Approximation of AP-SVM

The previous two subsections provide exact algorithms for loss-augmented inference that reduce the time require for training an AP-SVM. However, despite these improvements, AP-SVM might be slower to learn compared to simpler frameworks such as the binary SVM, which optimizes the surrogate 0-1 loss. The disadvantage of using the binary SVM is that, in general, the 0-1 loss is a poor approximation for the AP loss. However, the quality of the approximation is not uniformly poor for all samples, but depends heavily on their separability. Specifically, when the 0-1 loss of a set of samples is 0 (that is, they are linearly separable by a binary SVM), their AP loss is also 0. This observation inspires us to approximate the AP loss over the entire set of training samples using the AP loss over the subset of difficult samples. In this work, we define the subset of difficult samples as those that are incorrectly classified by a simple binary SVM.

Formally, given the complete input $\mathbf{X}$ and the ground-truth ranking matrix $\mathbf{R}^*$, we represent individual samples as $\mathbf{x}_i$ and their class as $y_i$. In other words, $y_i = 1$ if $i \in \mathcal{P}$ and $y_i = -1$ if $i \in \mathcal{N}$. In order to approximate the AP-SVM, we adopt a two stage strategy. In the first stage, we learn a binary SVM by minimizing the regularized convex upper bound on the 0-1 loss over the entire training set. Since the loss-augmented inference for 0-1 loss is very fast, the parameters $\mathbf{w}_0$ of the binary SVM can be estimated efficiently. We use the binary SVM to define the set of easy samples as $\mathbf{X}_e = \{\mathbf{x}_i, y_i \mathbf{w}_0^\top \phi_i(\mathbf{x}) \geq 1\}$. In other words, a positive sample is easy if it is assigned a score that is greater than 1 by the binary SVM. Similarly, a negative sample is easy if it is assigned a score that is less than -1 by the binary SVM. The remaining difficult samples are denoted by $\mathbf{X}_d = \mathbf{X} - \mathbf{X}_e$ and the corresponding ground-truth ranking matrix by $\mathbf{R}_d^*$. In the second stage, we approximate the AP loss over the entire set of samples $\mathbf{X}$ by the AP loss over the difficult samples $\mathbf{X}_d$ while ensuring that the samples $\mathbf{X}_e$ are correctly classified. In order to accomplish this, we solve the following optimization problem:

$$
\begin{aligned}
\min_{\mathbf{w}} \quad & \frac{1}{2}||\mathbf{w}||^2 + C\xi \\
\text{s.t.} \quad & \mathbf{w}^\top \Psi(\mathbf{X}_d, \mathbf{R}_d^*) - \mathbf{w}^\top \Psi(\mathbf{X}_d, \mathbf{R}_d) \geq \Delta(\mathbf{R}_d^*, \mathbf{R}_d) - \xi, \forall \mathbf{R}_d, \\
& y_i \left( \mathbf{w}^\top \phi(\mathbf{x}_i) \right) > 1, \forall \mathbf{x}_i \in \mathbf{X}_e.
\end{aligned}
\tag{10}
$$

In practice, we can choose to retain only the top $k\%$ of $\mathbf{X}_e$ ranked in descending order of their score and push the remaining samples into the difficult set $\mathbf{X}_d$. This gives the AP-SVM more flexibility to update the parameters at the cost of some additional computation.

## 4 Experiments

We demonstrate the efficacy of our methods, described in the previous section, on the challenging problems of action classification and object detection.

### 4.1 Action Classification

**Dataset.** We use the PASCAL VOC 2011 [7] action classification dataset for our experiments. This dataset consists of 4846 images, which include 10 different action classes. The dataset is divided into two parts: 3347 'trainval' person bounding boxes and 3363 'test' person bounding boxes. We use the 'trainval' bounding boxes for training since their ground-truth action classes are known. We evaluate the accuracy of the different instances of SSVM on the 'test' bounding boxes using the PASCAL evaluation server.

**Features.** We use the standard poselet [12] activation features to define the sample feature for each person bounding box. The feature vector consists of 2400 action poselet activations and 4 object detection scores. We refer the reader to [12] for details regarding the feature vector.

**Methods.** We present results on five different methods. First, the standard binary SVM, which optimizes the 0-1 loss. Second, the standard AP-SVM, which uses the inefficient loss-augmented inference described in Algorithm 1. Third, AP-SVM-SEARCH, which uses efficient search to compute the optimal interleaving rank for each negative sample using Algorithm 2. Fourth, AP-SVM-SELECT, which uses the selective ranking strategy outlined in Algorithm 3. Fifth, AP-SVM-APPX, which employs the approximate AP-SVM framework described in subsection 3.3. Note that, AP-SVM, AP-SVM-SEARCH and AP-SVM-SELECT are guaranteed to provide the same set of parameters since both efficient search and selective ranking are exact methods. The hyperparameters of all five methods are fixed using 5-fold cross-validation on the 'trainval' set.

**Results.** Table 1 shows the AP for the rankings obtained by the five methods for 'test' set. Note that AP-SVM (and therefore, AP-SVM-SEARCH and AP-SVM-SELECT) consistently outperforms binary SVM by optimizing a more appropriate loss function during training. The approximate AP-SVM-APPX provides comparable results to the exact AP-SVM formulations by optimizing the AP loss over difficult samples, while ensuring the correct classification of easy samples. The time required to compute the most violated rankings for each of the five methods in shown in Table 2. Note that all three methods described in this paper result in substantial improvement in training time. The overall time required for loss-augmented inference is reduced by a factor of $5 - 10$ compared to the original AP-SVM approach. It can also be observed that though each loss-augmented inference step for binary SVM is significantly more efficient than for AP-SVM (Table 3), in some cases we observe that we required more cutting plane iterations for binary SVM to converge. As a result, in some cases training binary SVM is slower than training AP-SVM with our proposed speed-ups.

| Object class | Binary SVM | AP-SVM | AP-SVM-APPX | | |
| --- | --- | --- | --- | --- | --- |
| | | | k=25% | k=50% | k=75% |
| Jumping | 52.580 | 55.230 | 54.660 | 55.640 | 54.570 |
| Phoning | 32.090 | 32.630 | 31.380 | 30.660 | 29.610 |
| Playing instrument | 35.210 | 41.180 | 40.510 | 38.650 | 37.260 |
| Reading | 27.410 | 26.600 | 27.100 | 25.530 | 24.980 |
| Riding bike | 72.240 | 81.060 | 80.660 | 79.950 | 78.660 |
| Running | 73.090 | 76.850 | 75.720 | 74.670 | 72.550 |
| Taking photo | 21.880 | 25.980 | 25.360 | 23.680 | 22.860 |
| Using computer | 30.620 | 32.050 | 32.460 | 32.810 | 32.840 |
| Walking | 54.400 | 57.090 | 57.380 | 57.430 | 55.790 |
| Riding horse | 79.820 | 83.290 | 83.650 | 83.560 | 82.390 |

Table 1: *Test* AP *for the different action classes of* PASCAL VOC *2011 action dataset. For* AP-SVM-APPX*, we report test results for 3 different values of k, which is the percentage of samples that are included in the easy set among all the samples that the binary* SVM *had classified with margin > 1.*

| Binary SVM | AP-SVM | AP-SVM-SEARCH | AP-SVM-SELECT | AP-SVM-APPX (K=50) | ALL |
|---|---|---|---|---|---|
| 0.1068 | 0.5660 | 0.0671 | 0.0404 | 0.2341 | 0.0251 |

Table 2: *Computation time (in seconds) for computing the most violated ranking when using the different methods. The reported time is averaged over the training for all the action classes.*

| Binary SVM | AP-SVM | AP-SVM-SEARCH | AP-SVM-SELECT | AP-SVM-APPX (K=50) | ALL |
|---|---|---|---|---|---|
| 0.637 | 13.192 | 1.565 | 0.942 | 8.217 | 0.689 |

Table 3: *Computation time (in milli-seconds) for computing the most violated ranking per iteration when using the different methods. The reported time is averaged over all training iterations and over all the action classes.*

### 4.2 Object Detection

**Dataset.** We use the PASCAL VOC 2007 [6] object detection dataset, which consists of a total of 9963 images. The dataset is divided into a 'trainval' set of 5011 images and a 'test' set of 4952 images. All the images are labelled to indicate the presence or absence of the instances of 20 different object categories. In addition, we are also provided with tight bounding boxes around the object instances, which we ignore during training and testing. Instead, we treat the location of the objects as a latent variable. In order to reduce the latent variable space, we use the selective-search algorithm [17] in its fast mode, which generates an average of 2000 candidate windows per image.

**Features.** For each of the candidate windows, we use a feature representation that is extracted from a trained Convolutional Neural Network (CNN). Specifically, we pass the image as input to the CNN and use the activation vector of the penultimate layer of the CNN as the feature vector. Inspired by the work of Girshick *et al.* [9], we use the CNN that is trained on the ImageNet dataset [4], by rescaling each candidate window to a fixed size of $224 \times 224$. The length of the resulting feature vector is $4096$.

**Methods.** We train latent AP-SVMs [1] as object detectors for 20 object categories. In our experiments, we determine the value of the hyperparameters using 5-fold cross-validation. During testing, we evaluate each candidate window generated by selective search, and use non-maxima suppression to prune highly overlapping detections.

**Results.** This experiment places high computational demands due to the size of the dataset (5011 'trainval' images), as well as the size of the latent space (2000 candidate windows per image). We compare the computational efficiency of the loss-augmented inference algorithm proposed in [19] and the exact methods proposed by us. The total time taken for loss-augmented inference during training, averaged over the all the 20 classes, is 0.3302 sec for our exact methods (SEARCH+SELECT) which is significantly better than the 6.237 sec taken by the algorithm used in [19].

## 5 Discussion

We proposed three complementary approaches to improve the efficiency of learning AP-SVM. The first two approaches exploit the problem structure to speed-up the computation of the most violated ranking using exact loss-augmented inference. The third approach provides an accurate approximation of AP-SVM, which facilitates the trade-off of test accuracy and training time.

As mentioned in the introduction, our approaches can also be used in conjunction with other learning frameworks, such as the popular deep convolutional neural networks. A combination of methods proposed in this paper and the speed-ups proposed in [10] may prove to be effective in such a framework. The efficacy of optimizing AP efficiently using other frameworks needs to be empirically evaluated. Another computational bottleneck of all SSVM frameworks is the computation of the joint feature vector. An interesting direction of future research would be to combine our approaches with those of sparse feature coding [3, 8, 18] to improve the speed to AP-SVM learning further.

## 6 Acknowledgement

This work is partially funded by the European Research Council under the European Community's Seventh Framework Programme (FP7/2007-2013)/ERC Grant agreement number 259112. Pritish is supported by the TCS Research Scholar Program.

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
