[Supplementary Material]

# Supplementary Material - Efficient Optimization for Average Precision SVM

**Pritish Mohapatra**
IIIT Hyderabad
pritish.mohapatra@research.iiit.ac.in

**C.V. Jawahar**
IIIT Hyderabad
jawahar@iiit.ac.in

**M. Pawan Kumar**
Ecole Centrale de Paris & INRIA Saclay
pawan.kumar@ecp.fr

## Appendix A

We provide a proof for proposition1. Before moving to the proposition, we first state the following lemmas, which easily lead to the proposition. For the sake of clarity of discussion, we will split the summand term in the summation $\delta_j(i)$ as follows:

$$\delta_j\,(i) \;=\; f_1\,(j,i) + f_2\,(j,i) \;=\; \sum_{k=i}^{|\mathcal{P}|} g_1\,(j,k) + \sum_{k=i}^{|\mathcal{P}|} g_2\,(j,k),$$

$$g_1\,(j,k) \;=\; \frac{1}{|\mathcal{P}|}\left(\frac{j}{j+k} - \frac{j-1}{j+k-1}\right), g_2\,(j,k) \;=\; -\frac{2\left(s_k^p - s_j^n\right)}{|\mathcal{P}||\mathcal{N}|}.$$

Please note that the functions $f_1(j,i)$ and $f_2(j,i)$ are cumulative sums of $g_1(j,i)$ and $g_2(j,i)$ respectively, in the decreasing direction of $i$. Therefore, for ease of reasoning, we shall analyse the trend of these functions in the decreasing direction of $i$.

**Lemma 1.** *For $k < j$, $g_1(j,k)$ monotonically decreases with decreasing $k$, that is $\forall\, k < j$ $g_1(j, k-1) \leq g_1(j,k)$.*

*Proof.* For $j \geq 1$ and $k \geq 1$, $(j+k) > j \Rightarrow j(j+k) - (j+k) < j(j+k) - j \Rightarrow \frac{j}{j+k} > \frac{j-1}{j+k-1}$. So, term $g_1(j,k) > 0$ for all $k \geq 1$. It can also be verified that the function $g_1(j,k)$ is 0 at 0 and has a single maxima for $k \in \Re^+$, at $k = \sqrt{j(j-1)}$. From this we can conclude that for discrete $k \in \mathbb{Z}^+$, $g_1(j,k)$ would have maximum value either at $k = j$ or $k = j-1$. Therefore, for $k < j$, $g_1(j,k)$ would monotonically decrease with decreasing $k$. $\square$

**Lemma 2.** *For $k < j$, $g_2(j,k)$ monotonically decreases with decreasing $k$, that is $\forall\, k < j$ $g_2(j, k-1) \leq g_2(j,k)$.*

*Proof.* In $g_2(j,k)$, the negative score $s_j^n$ is a constant for a given $j$. Whereas, the positive scores $s_k^p$ being sorted in descending order, monotonically increase as $k$ decreases. Therefore, $g_2(j,k)$ which is $-s_k^p + constant$, monotonically decreases as $k$ decreases. $\square$

**Proposition 1.** *The discrete function $\delta_j(i)$, defined in equation-5 of the main text, is unimodal in the domain $\{1, \cdots, p\}$, where $p = \min\{|\mathcal{P}|, j\}$.*

*Proof.* From lemmas 1 and 2, for $k < j$, $g_1(j,k)$ and $g_2(j,k)$ monotonically decreases with decreasing $k$. As a result, $g_1(j,k) + g_2(j,k)$ also monotonically decreases when $k$ is decreased from right to left of the number line. Here, there can be 3 scenarios,

$(i)$ $(g_1(j,1) + g_2(j,1)) \geq 0$. In this case, as the function is monotonic and decreases towards left,

$$
\begin{aligned}
& (g_1(j,i) + g_2(j,i)) \geq 0, for\ i \in \{1,2,...,j\} \\
\Rightarrow \quad & \delta_j(i) - \delta_j(i+1) \geq 0, for\ i \in \{1,2,...,\} \\
\Rightarrow \quad & \delta_j(i) \geq \delta_j(i+1), for\ i \in \{1,2,...,\}
\end{aligned}
$$

Therefore, according to definition of unimodality, $\delta_j(i)$ would be unimodal with $k = 1$.

$(ii)$ $(g_1(j,j-1) + g_2(j,j-1)) \leq 0$. In this case, using similar reasoning as above,

$$
\begin{aligned}
& (g_1(j,i) + g_2(j,i)) \leq 0, for\ i \in \{j-1,...,1\} \\
\Rightarrow \quad & \delta_j(i) - \delta_j(i+1) \leq 0, for\ i \in \{j-1,...,1\} \\
\Rightarrow \quad & \delta_j(i) \leq \delta_j(i+1), for\ i \in \{j-1,...,1\}
\end{aligned}
$$

Therefore, $\delta_j(i)$ would be unimodal with $k = j - 1$.

$(iii)$ $(g_1(j,1) + g_2(j,1)) \leq 0$ and $(g_1(j,j-1) + g_2(j,j-1)) \geq 0$. In this case, there should exist a point across which the function $(g_1 + g_2)$ changes its sign from positive to negative when moving from right to left. In other words, there should exist $k \in 1, 2, \ldots, j-1$, such that,

$$
\begin{aligned}
& (g_1(j,i) + g_2(j,i)) \geq 0, i \in \{k+1,...,j\} \\
& (g_1(j,i) + g_2(j,i)) \leq 0, i \in \{1,...,k\} \\
\Rightarrow \quad & \delta_j(i) - \delta_j(i+1) \geq 0, for\ i \in \{k,...,j-1\} \\
& \delta_j(i) - \delta_j(i+1) \leq 0, for\ i \in \{j-1,...,1\} \\
\Rightarrow \quad & \delta_j(i) \geq \delta_j(i+1), for\ i \in \{k,...,j-1\} \\
& \delta_j(i) \leq \delta_j(i+1), for\ i \in \{j-1,...,1\}
\end{aligned}
$$

Here too, $\delta_j(i)$ satisfies the conditions for unimodality with $k$ being the maximum point.

In all the 3 of the exhaustive cases, $\delta_j(i)$ satisfies the conditions for unimodality. Hence, $\delta_j(i)$ is unimodal in the region $\{1, 2, \ldots, j-1\}$. As a function which is unimodal in a certain region would also be unimodal in a subset of the region, $\delta_j(i)$ is unimodal in the region $\{1, 2, \ldots, p\}$, where, $p = \min(|\mathcal{P}|, j)$. $\qquad \square$

## Appendix B

Here we report some more results which helps in analysing the effect of varying the number of positive and negative training samples. We perform experiments in which we vary the number of positives and negatives training samples for the action class 'phoning'. As can be seen in Fig. 1 the time required to perform loss-augmented inference is significantly lower for our methods. The computation time for the loss augmented inference in AP-SVM-SEARCH is more stable as for large negative set sizes the improvement is always guaranteed.

## Appendix C

For our object detection experiments, we report the detection AP for all the 20 object categories obtained by latent AP-SVM as well as by the standard latent SVM which is used as a baseline. For all object categories other than 'bottle', latent AP-SVM does better than latent SVM on the test set. For 15 of the 20 object categories, we get statistically significant improvement with latent AP-SVM over latent SVM (using paired t-test with p-value less than 0.05). While latent AP-SVM gives an overall improvement of 7.12% compared to latent SVM, for 5 classes it gives an improvement of more than 10%. The bottom 2 classes with the least improvement obtained by latent AP-SVM, 'chair' and 'bottle' seem to be difficult object categories to detect, with detectors registering very low detection APs. In conjunction with the overall superior performance of latent AP-SVM, the efficient method proposed by this paper makes a good case for optimizing AP loss rather than 0-1 loss for tasks like object detection.

Figure 1: *Computation time for solving all the loss augmented inference problems during the complete training of the* SVM*s, while the no. of total, negative and positive samples are varied.*

| Object category | latent SVM | latent AP-SVM |
|---|---|---|
| Aeroplane | 46.60 | 48.18 |
| Bicycle | 48.53 | 61.45 |
| Bird | 33.31 | 36.73 |
| Boat | 15.23 | 19.66 |
| Bottle | 6.10 | 1.01 |
| Bus | 37.01 | 49.51 |
| Car | 61.28 | 66.78 |
| Cat | 38.12 | 40.77 |
| Chair | 2.71 | 3.23 |
| Cow | 21.06 | 38.52 |
| Dining-table | 14.20 | 39.53 |
| Dog | 33.55 | 36.25 |
| Horse | 46.14 | 53.86 |
| Motorbike | 29.97 | 34.81 |
| Person | 29.58 | 30.41 |
| Potted-plant | 21.27 | 23.03 |
| Sheep | 11.65 | 32.20 |
| Sofa | 36.66 | 42.03 |
| Train | 29.71 | 37.10 |
| TV-monitor | 27.31 | 37.26 |

Table 1: Object category wise detection AP (%) on PASCAL VOC 2007 test set.