[Reviews · NeurIPS 2014]

Submitted by Assigned_Reviewer_37

This paper proposed three schemes to speed up the AP-SVM algorithm originally proposed in [12]. All the schemes are very interesting. They can be very useful for solving larger-scale AP-SVM problems.

The results are somewhat not so comprehensive.

1) It would be interesting to see a speed-up result of the combination of all three tricks.
2) It would be interesting to see the demonstration of the algorithm performing on large-scale ranking tasks.

On the table 2, if the percentage (e.g., k=75%) is the portion of hard examples, would it k=75% be the case that is closest to AP-SVM? But the results seems to indicate that k=25% is the closest to AP-SVM.
Summary: An interesting paper but the results are not so comprehensive.

Submitted by Assigned_Reviewer_42

This paper proposes three different approaches to speed up the learning for AP-SVM (a structured SVM with AP loss).
1. The first approach uses the unimodal behavior of the loss function, so uses binary search to speed up calculating the loss.
2. The second approach notices that usually we have more negative samples compared to positive ones, and in calculating the loss, relative rank of negative samples falling between two consequent positive ones is not important. Hence, it speeds up by identifying all negatives between two consequent positive ones and assigns the same rank to all of them.
3. finally, it approximates the training by using 0-1 loss for easy examples (easy under 0-1 loss SVM).

In my view, all three tricks are interesting, clever, and effective in speeding up the training. They are described clearly.

In working with big data, speeding up the training is an important problem, so NIPS audience are interested in this topic.

I think the evaluation can be more thorough. Particularly, it might be important to try the algorithm on a different problem and also with larger number of training data. For instance, as I understood, the number of negatives in this task is not that large. There are other tasks like object detection that any possible bounding box in the image is considered a negative sample, so the number of negative samples can be exponentially large. I would suggest applying this algorithm on one of those tasks to show off the real power of this algorithm.

In Table 1, I was thinking AP-SVM-SELECT should be faster than AP-SVM-SEARCH, but it is not the case. It might be the case in another experiment with larger number of examples.

Also, in Fig 2-right, I was thinking by increasing the number of positive examples, AP-SVM-SELECT should loose its advantage and become slower than AP-SVM-SEARCH since the gaps become small, but it is not the case in this Fig.

In Fig 2 middle, I am surprised that the binary SVM is slower than the AP-SVM-SEARCH. I thought the only difference is in calculating the loss, and calculating 0-1 loss is trivial. Maybe I didn't understand it completely.

I think these should be discussed and explained if possible in the text.

Is it possible to combine the first two tricks? Or even combine all three? What do we expect to see? Line 21 in abstract says they are complementary, so should be nice to show it.

Minor points:
Line 431, "hard" should change to "easy" to be consistent with Line 344. Am I right?
Line 334, in "is less than 1", I think "1" should change to "-1".
Summary: I like the main idea of this work. My only problem is with the evaluation and discussion, so the rebuttal may be helpful.

Submitted by Assigned_Reviewer_44

In this paper, the authors propose several computational optimizations of the existing algorithm AP-SVM [Yue et al.]. The major drawback of this algorithm is the computational cost of ensuring the constraint of the optimization problem. The authors propose two exact methods (i.e. that give the same result as AP-SVM), the first one based on improving the search of the most violated constraint, the second one using the fact that the negative samples only need to be ranked relatively to positive ones. The approximate method relies on the fact that some samples are easy to classify with a binary SVM and the AP-SVM can be optimized only on “difficult samples”, which leads to a two-step procedure for inference, but reduce drastically the computational cost of the optimization.

The reading of this paper suffers from the fact that the authors rely a lot on the paper from Yue et al. and don’t fully introduce all the definitions and the context. For instance, the notation \Delta(R^*, R) in eq. (3) and (4) is not defined.
The proposed computational optimizations are correctly described and justified. The fact that the two first methods are exact is correctly explained. The approximation made for the last method seems reasonable and the experiments are convincing.

Overall, I am convinced that the computational improvements are significant while error performance stays equals or comparable to the initial algorithm.
However, I think that the scope of this paper is really restricted, as methods like LambdaMART or RankNet are really more popular than the Ranking SVMs.
Summary: The quality of the paper and of the results is good, but the scope of the contribution is restricted.
Author Feedback
Author rebuttal: We thank the reviewers for their insightful comments, which will help improve the quality of the paper.

Our paper proposes three novel ideas to speed-up loss-augmented inference for training AP-SVM [12]. The first, SEARCH, is guaranteed to obtain an asymptotic decrease in the computational complexity compared to the method of [12]. The second, SELECT, uses the fact that many negative samples can share the same interleaving rank and thus avoids the expensive sorting step of [12]. While its complexity is higher than [12] in the worst case, in practice this strategy results in a speed-up. The third, APPX, approximates the AP loss with a combination of the 0-1 loss over easy samples and the AP loss over the hard samples. Experiments on the VOC 2010 action classification dataset confirm that our ideas result in a significant speed-up.

All the reviewers agree that the proposed ideas are interesting and effective, and that the paper describes them clearly. A common suggestion from the reviewers is to show more empirical evidence, which we provide below. We also provide answer the specific questions raised in the reviews.

Q: Results for combining all ideas? (AR37, AR42)

The timing results for all three ideas individually, and their combination, for the VOC action classification dataset are as follows:
[12]: 0.5660 sec
SEARCH: 0.0671 sec
SELECT: 0.0404 sec
APPX (k=50%): 0.2341 sec
ALL: 0.0251 sec

Please note that the SELECT results have improved considerably. This is achieved by restricting the SELECT strategy to negative samples with interleaving rank = P (those ranked below all positives). In our experiments, we found that in later iterations of the optimization, a large number of negative samples have interleaving rank = P, while the negative that share other values of interleaving ranks decrease considerably.

As can be seen, the three ideas are complementary since the time for ALL is less than the time for the individual ideas.

Q: Results for a large-scale problem? (AR37, AR42)

As suggested by the reviewers, we evaluated our ideas for the task of object detection using latent AP-SVM on the VOC 2007 detection dataset. We used the features and hyperparameter values provided by the authors of the following publication:
http://cvit.iiit.ac.in/projects/lapsvm/Research_files/lapsvm_journal.pdf
Due to time constraints, we only tested the exact methods, SEARCH and SELECT, for which no further cross-validation was required. In the limited time, we could complete experiments for 15 of the 20 classes. The timing results, averaged over the 15 classes, are as follows:
[12]: 6.006 sec
SEARCH+SELECT: 0.3180 sec

We will include the new results in the paper to provide stronger support for our ideas.

Q: Other ranking algorithms? (AR44)

The following survey paper reports comparable performance for LambdaMART and ranking SVMs:
Busa-Fekete, R., et al. "An apple-to-apple comparison of Learning-to-rank algorithms in terms of Normalized Discounted Cumulative Gain." ECAI 2012-20th European Conference on Artificial Intelligence. Vol. 242. 2012.

Furthermore, our speed-ups are not only applicable to AP-SVM, they can be used with any ranker that minimizes the upper bound on the AP loss. One such scenario is mentioned in the discussion section of our paper. Specifically, we can replace the final layer of the highly successful convolutional neural network such that we minimize the AP loss instead of the softmax loss. In this case, backpropagation requires solving the loss-augmented inference problem to compute the subgradient, which can be significantly speeded-up using our ideas.

Q: Performance of SELECT vs. SEARCH (AR42)

The effectiveness of SELECT depends heavily on how many negative samples share the same interleaving rank. As mentioned above, we observed that the number of negative samples with interleaving rank = P increases substantially with the iterations, while it decreases for the other interleaving rank values. By restricting ourselves to apply SELECT for only the value of P, we were able to substantially improve the performance of SELECT.

The results shown above indicate that the improved SELECT strategy is more effective than SEARCH for the action classification dataset.
As suggested by the reviewer, we will include our observations about the SELECT strategy in the paper.

Q: Binary SVM slower than SEARCH? (AR42)

While each loss-augmented inference step for binary SVM is significantly more efficient than for AP-SVM, in some cases we observed that we required more cutting plane iterations for binary SVM to converge. This is the reason why sometimes binary SVM is slower than training AP-SVM with our proposed speed-ups. We will report the number of iterations and the average time per iteration in the paper to clarify this point.

Q: 'k' in table 2, line 431? (AR37, AR42)

The reviewers correctly point out that k is the percentage of easy samples.

Q: Mistake in line 334? (AR42)

The reviewer correctly points out that this should be "less than -1" instead of "less than 1". We will correct this and the above mistake in the new version of the paper.